# Re-Thinking Ethics and Politics in Suicide Prevention: Bringing Narrative Ideas into Dialogue with Critical Suicide Studies

**DOI:** 10.3390/ijerph16183236

**Published:** 2019-09-04

**Authors:** Jennifer White, Jonathan Morris

**Affiliations:** School of Child and Youth Care, University of Victoria, P.O. Box 1700, STN CSC Victoria, BC V8W 2Y2, Canada

**Keywords:** critical suicide studies, narrative therapy, ethics, politics

## Abstract

The purpose of this paper is to explore the conviviality between practices of narrative therapy and the emerging field of critical suicide studies. Bringing together ideas from narrative therapy and critical suicide studies allows us to analyze current suicide prevention practices from a new vantage point and offers us the chance to consider how narrative therapy might be applied in new and different contexts, thus extending narrative therapy’s potential and possibilities. We expose some of the thin, singular, biomedical descriptions of the problem of suicide that are currently in circulation and attend to the potential effects on distressed persons, communities, and therapists/practitioners who are all operating under the influence of these dominant understandings. We identify some cracks in the dominant storyline to enable alternative descriptions and subjugated knowledges to emerge in order to bring our suicide prevention practices more into alignment with a de-colonizing, social justice orientation.

## 1. Introduction

Suicide rates are either increasing or continuing unabated in many parts of the world. It is clear that narrow, risk factor-based approaches to suicide prevention are limited. We argue that they over-emphasize individual traits and psychological qualities, and neglect to examine the ways that history, contexts, policies, discourses, and structures contribute to vulnerability, hopelessness, and distress. The purpose of this paper is to explore the conviviality between practices of narrative therapy and the emerging field of critical suicide studies. Drawing on ideas from narrative therapy to engage with this nascent field of suicide studies allows us to analyze current suicide prevention practices from a new vantage point and offers us the chance to consider how narrative therapy might be applied in new and different contexts, thus extending narrative therapy’s potential and possibilities.

In the sections that follow, we introduce ourselves and provide the context for our current interest in exploring alternatives to the dominant approach to studying and responding to suicide. We articulate our theoretical position, which is strongly influenced by critical, post-structural, and post-humanist approaches to scholarly inquiry and praxis. In keeping with a critically reflexive orientation, we write ourselves into the text, unsettling any assumptions about a value-free, neutral, or purely objective account of this work. This is a view from somewhere. We work from the assumption that suicide prevention is a social practice—or assemblage—that comprises bodies, identities, technologies, discourses, institutional artifacts, and cultural practices that are constantly shaping and re-shaping what can be said, thought, and done. We expose some of the thin, singular, biomedical descriptions of the problem of suicide that are currently in circulation and attend to the potential effects on distressed persons, communities, and therapists/practitioners who are all operating under the influence of these dominant understandings. We briefly introduce narrative therapy and community work as one potential alternative. This is a way of working that challenges familiar psychological understandings of persons and problems, which often start from the assumption that problems like suicide reside inside persons. Narrative therapy is based on the simple principle that problems are separate from people and thus the therapeutic (or community or pedagogical) task is to elicit, link, and circulate new stories, discourses, and cultural practices that are in keeping with the individual’s or group’s preferred future. In the final section we identify some cracks in the dominant storyline of suicide and suicide prevention to enable alternative descriptions and subjugated knowledges to take hold. In other words, we follow the pattern of a narrative therapy interview to show how power is never absolute, identity conclusions and claims about reality are not final, and ways of being in the world, including being suicidal, being ambivalent about living, or being hopeful, are fluid and emergent, contextually situated, and co-constituted with others through discourses, institutional practices, relational processes, and technologies.

We begin by situating ourselves as settlers on the unceded territories of the Coast Salish peoples on the west coast of Canada. More specifically, one of us (Jennifer) is a white settler who lives and works on the traditional territory of the Lkwungen and Songhees peoples and one of us (Jonathan) is a racialized settler living and working on the traditional territories of the Musqueam, Tsleil-Waututh, and Squamish peoples. By acknowledging whose territories we are on and taking some time to situate ourselves in relation to this colonial history, we are doing a number of things. First, we are reminding ourselves that our nation came into being through the theft of Indigenous peoples’ land, disavowal of Indigenous sovereignty, practices of genocide, and the violent removal of children from their families and communities [1]. Second, these territorial acknowledgements call on us to reckon with our privileges to see the specific ways that we have benefitted from the well-documented racist and structural arrangements that persist to this day. They also nudge us towards taking specific actions, including making meaningful reparations. Third, this way of situating ourselves, and acknowledging the painful legacies of these histories, is especially relevant for us as we attempt to re-think suicide outside of a narrow psychological or biomedical framework and bring our suicide prevention practices more into alignment with a de-colonizing, social justice orientation.

We also bring decades of experience to this discussion. For example, one of us (Jennifer) has worked as a suicide prevention counsellor, educator, policy consultant, and researcher and brings over 30 years of experience to this work. One of her earliest memories of working in a professional capacity under the influence of dominant ideas about suicide prevention was when she was working in a large residential treatment centre for children and youth who had a range of emotional and behavioural challenges as well as complex trauma histories. There had been a recent suicide by hanging in the facility and a new set of policies had been developed in response, ostensibly with the goal of reducing future deaths like this. The new policy dictated that, when children were placed in locked confinement rooms (which was usually the result of aggression or threatened violence), the protocol was to ensure that we, the child and youth care professionals, removed all belts, shoelaces, and any other potential materials that could be used as a ligature. The policy also stated that there were to be constant checks on the young person through both mounted cameras on the ceiling and via the small window on the bolted door. Surveillance and control were the primary responses to concerns about suicide. No questions were raised about the ethics of restraining or locking children up, or the trauma that we might be re-producing by doing so. The overall orientation was highly technical and procedural in nature, as evidenced by the fact that we were to document that we had followed through on all of the rules (presumably to avoid blame). Just as Chapman [2] describes how he “became a perpetrator” by participating in the confinement and restraint of children with disabilities, Jennifer recognizes that she was also actively complicit with perpetuating harm under the guise of ‘helping’ and ‘preventing suicide’. This devastating realization is something that haunts her to this day and serves as a reminder that ‘suicide prevention’—at any cost—can sometimes be coercive in nature and should never be assumed to be an unqualified good. The prevention imperative that is inexorably embedded within the study of suicide itself, has rightly been called into question by Tack [3]. We discuss this in more detail in a later section.

Jonathan has practiced in the field of suicide prevention for over 20 years, starting out as a youth volunteer on a national crisis line service at 16 years-old. During his career, Jonathan has worked as a youth suicide prevention educator, community developer, researcher, policy-maker, and non-profit leader. Jonathan’s first training in suicide prevention was founded on a non-interventionist approach, where the provision of empathic and non-judgemental care, even if someone was at the point of taking their own life, was paramount. Since then, Jonathan has had extensive involvement in the delivery of suicide prevention curricula for youth. While undertaking graduate studies, Jonathan started to question if contemporary suicide prevention education and its emphasis on individual intervention might be missing the broader social, cultural, and political context in which suicide sits, and additional opportunities for creating conditions that promote life and living.

### 1.1. Context for Concern

Despite the proliferation of evidence-based treatments, suicide prevention campaigns, toolkits, and public awareness strategies, suicide rates stubbornly persist, and have actually increased in some parts of the world, including the United States [4,5]. We live in the colonial state of Canada, where Indigenous (i.e., First Nations, Inuit, and Métis) rates of suicide are disproportionately higher, relative to the general population [6,7]. It is also well documented that many gay, lesbian, transgender, queer and non-binary (GLBTQ+) persons, particularly those who experience additional social inequities along the axes of race, class, education, have high rates of suicidal behavior, compared to their heterosexual counterparts [8]. Rates of suicide among girls and women are increasing in Canada and elsewhere, especially among those who have a history of childhood maltreatment, including sexual abuse [9,10]. Males, especially military veterans, the unemployed, gender and sexual minorities, and those living in rural areas, kill themselves more often than females across all age groups [11,12]. Those who live in socioeconomically deprived areas, and who are facing economic hardship, have also been shown to be at elevated risk for suicide [13].

In most parts of the world, more men kill themselves than women, and more women engage in suicidal behavior. At the same time, a critically reflexive approach to studying suicide shows how suicide statistics themselves are products of social processes that are highly gendered and influenced by local contexts and understandings about suicide (as opposed to neutral and objective facts) [14,15]. While suicide and self-harming behaviours “directly engage the social, psychological, and biological” [13] (p. 3), most research into suicide exists within narrow disciplinary frames that draw heavily from quantitative, positivist methodologies, and this is as true for sociology as it is for psychology [13,16]. Sociopolitical influences and ideological regimes (e.g., neoliberalism, austerity policies, colonialism, heteropatriarchy), strongly determine how social inequities will be experienced. These social forces also have a role to play in how shame, failure, and loss are experienced, and are relevant considerations when theorizing suicide and suicidal behaviours [13,17]. Neoliberal policies, colonial practices, and capitalist logics are typically not catalogued as suicide risk factors, and typically remain outside the scope of death investigation practices and empirical investigations into suicide. A rare exception is China Mills’ “psychopolitical autopsy” study of “austerity suicides” in the UK [18]. Within much suicidology research, it is individuals that get tagged as ‘high-risk’ and the focus then turns toward their individual psyches as the proper site for intervention.

This re-articulation of local expressions of injustice and social inequality into psychic suffering … is perceptible in the discourses that circulate around other marginalized and vulnerable groups that are more likely to die by suicide such as Indigenous people, the unemployed, those from sexually diverse populations and those in the criminal justice system [19] (p. 185).

Working against such a narrow biomedical reading of suicide, we take the position that elevated suicide rates among certain groups provides us with an opportunity to address the social and political contexts of suicide [13,18,19] to expose some of “the social and cultural conditions that inform these higher rates in the choice for death … so as to render certain bodies unintelligible, with very real material and practical consequences” [3] (p. 58). Meanwhile, increasing rates of suicide among white American men—who typically enjoy the greatest number of unearned benefits and privileges—have recently been linked to white fragility, racial anxieties, and the politics of racial resentment [20]. Clearly suicide cannot be explained through simple cause–effect reasoning and there are many paradoxes and contradictions that need to be accounted for when attempting to understand elevated rates of suicide among different groups. Intersectional analyses that recognize the uneven and shifting distribution of power and privileges along multiple social locations (i.e., gender, sexual orientation, ethnicity, ability, age, etc.) are likely to be more useful than static or essentialist readings of identity and suicidal subjectivities [8,20]. While we acknowledge that there has been a strong emphasis placed on individual, biomedical, and psychological explanations of suicide in the modern era, and we seek to broaden the frames for understanding and responding to suicide, we do not mean to suggest that a singular approach (psychological or structural) can account for suicide’s complexity. Our intention is to articulate a theoretically grounded, ethically attuned, and practical approach to understanding and responding to suicide. It is not meant to be a panacea, but could be considered as one possibility among others.

### 1.2. Theoretical Framework

In keeping with our narrative therapy orientation, we draw from a post-structural theoretical framework which emphasizes the role of culture, language, and discourse in understanding persons, problems, and lived realities [21,22]. Post-structuralism, exemplified in the work of Foucault and other French theorists, “challenges the rationalism and realism that underlies structuralism’s faith in scientific method, in progress, and in discerning and identifying universal structures of all cultures and the human mind… it is suspicious of meta-narratives, transcendental arguments,, and final vocabularies” [22] (p. 131, emphasis in original). Informed by a post-structuralist understanding, we eschew a singular, stable, essentialist, or universal construction of the self and instead work from the premise that we are all discursively constituted, constantly emerging in, and through, our relations with others. Multiple stories and discourses constitute our identities in culturally specific ways and we not locked into rigid, problem-saturated identity categories, such as “depressed” or “mentally ill” or “at-risk” or “suicidal” [21]. Even in the most oppressive of circumstances, there is always the possibility for agency, resistance, and novel responses, which inevitably requires the presence of others. In other words, we are always operating within a set of ongoing and emerging relations that are stabilized or disrupted through socially coordinated actions over time [23].

Such a theoretical understanding has consequential implications for how we think about suicide and our responses to it. Specifically, suicide is not something that is located inside persons—“a compulsory ontology of pathology” [24] (p. 31)—but rather it is a historically contingent, contextually nuanced, social phenomenon characterized by multiplicity, instability, and ineffability. There are aspects of suicide that can never be fully known, understood, or captured by the techniques of science, despite the bold claims of many mainstream suicidologists [25]. It is this very instability that makes the problem of suicide conducive to multiple interpretations, which in turn creates space for alternative meanings and understandings. We, and others, can always be more than our suicidal acts, and this is perhaps especially important to remember when working with survivors who have lost a loved one to suicide [26].

We also take up a relational notion of ethics [23], which departs from codified, rule-based, approaches to professional ethics that often locate the ‘doing of ethics’ inside the head of the practitioner. A relational approach to ethics focuses on “what people do together and what their ‘doing’ makes” [27] (p. 425). In this view of ethics, what is determined to be right or good emerges through fluid, socially coordinated actions, within specific social, historical, and cultural contexts. When we are acting ethically, we are acting with others and are called to recognize our deep interdependency and implication in a shared future. A high tolerance for uncertainty is required as we co-consitute ourselves with others through joint learning and action, curiosity, and mutual discovery. Ethical questions shift away from asking “what is the correct ethical principle to apply in this situation?” towards “who are we becoming?” and “what possibiliites are being crafted?” [27] (p. 429).

It is also worth elaborating on our understanding of critique, which is informed by Foucault who states:
I don’t think that criticism can be set against transformation, “ideal” criticism against “real” transformation … A critique does not consist in saying that things aren’t good the way they are. It consists in seeing on what types of assumptions, of familiar notions, or established, unexamined ways of thinking, the accepted practices are based.
… *There is always a little thought occurring in the most stupid institutions; there is always thought even in silent habits … Criticism consists in uncovering that thought and trying to change it; showing that things are not as obvious as people believe, making it so that what is taken for granted is no longer taken for granted. To do criticism is to make harder those acts which are now too easy … Understood in these terms, criticism (and radical criticism) is utterly indispensable for any transformation.*
… *as soon as people begin to have trouble thinking things the way they have been thought, transformation becomes at the same time very urgent, very difficult, and entirely possible*.[27] (pp. 456–457)


In keeping with Foucault’s framing of criticism, both of us actively resist saying or implying that current approaches to suicide prevention “aren’t good the way they are” [27] and should be replaced with a new and dominant doctrine of practices informed by critical suicide studies. They would inadvertently re-create the very structures we hope to trouble and interrogate. Rather, both of us, alongside scholars, researchers, practitioners, and people with direct experience of suicide around the world, are finding ways to make visible the taken for granted assumptions governing the practice of suicide prevention and research, to bring increased reflexivity, creativity and flexibility to the field. To do this, we have engaged with narrative ways of knowing, doing, and being to invite opportunities for individuals and communities to “begin to have trouble thinking things the ways they have been thought” [27] and to invite expanded possibilities in relation to suicide.

### 1.3. Dominant Constructions of Suicide (Prevention)

The vocabulary of suicidology and suicide prevention reveals its rationalist premises and positivist assumptions. For example, we routinely speak about evidence-based practices, risk management, assessment and treatment, monitoring, expert interventions, and scientific approaches to the study and prevention of suicide. While none of these approaches are inherently wrong, they are all generally grounded in an expert, biomedical framework and thus tend to narrow the range of possibilities for thinking about and responding to suicide. Most suicide prevention practices continue to be based on individual-level theorizing which in turn generate solutions that emphasize individual responsibility, self-monitoring, and seeking expert assistance. Meanwhile, philosophical, ethical, political, and spiritual perspectives on suicide, which have complex and unique histories of their own, have largely disappeared from discussions and debates within mainstream suicidology [19]. As Bracken et al. [28] (p. 433) put it: “All forms of suffering involve layers of personal history, embedded in a nexus of meaningful relationships that are, in turn, embedded in cultural and political systems.”

Neoliberal discourses of risk and responsibility typically locate the problem of suicide (and depression) inside individual persons in a de-politicized way, placing the responsibility (onus) for change on individuals through therapeutic techniques of rational problem-solving, self-management, coping, and skill-building. Meanwhile, much of the mainstream suicidology research seems to be about taming suicide through a push for unification and standardization [29]. This is visible through the emphasis on evidence-based strategies and standardized protocols which tend to smooth over differences, flux, contradictions, and contexts. Normative (often unspoken) understandings of mental health circulate through professional therapeutic discourses and “public pedagogies of mental health,” that urge people to ‘reach out’ and ‘talk about it’ [30].

Contradictory ideas regarding ‘responsibility for suicide’ also circulate within everyday suicide prevention practices, whereby suicidal persons are deemed both not responsible for their own deaths as a result of their mental illnesses (which ostensibly impair their decision-making abilities), and also ultimately responsible, which has the effect of deflecting blame away from other persons, relations, or social arrangements that may be implicated in the emergence of hopelessness and suicidality. Meanwhile community citizens, peers, and other gatekeepers are trained to ‘be responsible’ by learning the facts about suicide and how to help. This typically involves referring suicidal people to mental health services, where they will typically encounter practices of assessment, surveillance and risk management, which may not always be experienced as helpful. For example, a recent qualitative study demonstrated that suicidal persons who access psychotherapy often conceal their suicidality for fear of being involuntarily hospitalized [31]. This is an important finding as it shows how standard suicide prevention practices (i.e., risk assessment, surveillance, monitoring, involuntary hospitalization), can potentially jeopardize the cultivation of trust and may interfere with an honest and open dialogue, which are the foundations upon which any therapeutic relationship rests [32]. Furthermore, possibilities for more informal types of friendship and support, based on unconditional acceptance and reciprocity, may actually be undermined when too much emphasis is placed on ensuring suicidal persons access professional help, to the exclusion of other relational possibilities [19]. Such concerns are also consistent with recent questions raised by Tack [3], who shows how the prevention imperative that is unproblematically and inexorably embedded within all research about suicide—including critical suicide studies—starts from a taken-for-granted assumption that all life is to be lived and that living is the natural state of all humans.

This line of thinking resonates with critical public health scholars who consider the current western preoccupation with “health,” and its existence within moralizing rhetoric, to be ripe for interrogation [33]. Specifically, they take a questioning stance towards the taken-for-granted acceptance of health as an unqualified good or singularity and argue that health is a desired state, a prescribed state as well as an ideological position [33] (p. 2). We could also argue that “suicide prevention” is also generally assumed to be an unequivocally desirable value “… that one cannot but choose” [33] (p. 15). After all, who could be against suicide prevention? And yet when we consider that “suicide prevention” is not a unitary, ahistorical, or natural position but rather a set of ongoing practices that are deeply entangled with specific ideologies, histories, institutional practices, discourses and politics that reflect a range of interests (including government, pharmaceutical companies, medicine, training organizations, media, etc.), we begin to appreciate the important work that critical theorizing in suicide prevention can do [3]. For critically oriented suicide studies scholars, this means raising questions about suicide prevention so that we can see the ways that it holds multiple and competing interests and agendas, while simultaneously concealing other social realities and arrangements. In short, by introducing a “stutter” [34] into assumptions governing the study and practice of suicide and suicide prevention, we are given space to think and act anew.

## 2. Critical Suicide Studies

Given growing dissatisfaction with biomedical framings of suicide, and an over reliance on individualistic approaches to suicide prevention—that omit or take for granted the social, historical, and political contexts of peoples’ lives—many scholars, practitioners, and policy leaders are looking for alternatives [3,13,14,15,16,17,18,19,24,35]. By drawing on multiple, embodied knowledges and intellectual traditions, the emerging field of critical suicide studies seeks to build upon and challenge some of the taken-for-granted positivist and biomedical logics that have come to dominate the study of suicide and the practice of suicide prevention in recent decades.

Like practices of narrative therapy, critical suicide studies is not a singular or monolithic entity. It comprises many strands, that are in constant states of emergence, but tends to cohere around a few key ideas. It is typically critical of universal, generalizable, expert, totalizing knowledges (the already known and represented). It often challenges singular, essentialist identities and does not assume a unitary self [35]. The field is in its infancy, but nonetheless, it seeks to be critically reflexive and is willing to turn the critical gaze on itself. Critical suicide studies questions taken-for-granted assumptions about what suicide is and situates distress in sociopolitical, cultural, and historical contexts and discourses. It is explicitly ethical and political in its orientation with a focus on decolonization, social justice, equity, and inclusion, and aims to create a world worth living in. It seeks to create the conditions for interdependent and collective life where all can flourish. In other words, “Critical suicidology works in the midst of contradictions, seeks to enable multiple approaches to proliferate, and attempts to ask fresh new (perhaps previously unthinkable and unsayable) questions in an effort to think outside of the received view of practice” [35] (p. 7).

When we begin to recognize suicide’s multiplicity, irreducibility, complexities and contradictions, we begin to appreciate the need for multiple approaches and responses. The field of critical suicide studies is not looking to replace one universal or standardized approach with another. Nor is it meant to discredit or de-value the important and compassionate work being undertaken by researchers, professional practitioners, volunteers, and family members who are committed to better understanding suicide and who are engaged in the often difficult work of helping another person stay tethered to life. Rather, it is a movement that explicitly rejects over-simplifying what are highly complex, contingent social relations and histories. This implies moving away from intervening upon persons (i.e., relying on a ‘tick-box mentality’ to assess suicide risk) towards living and working alongside others to imagine and create alternative futures, positions, and ways of living together. Narrative therapy offers a philosophy, a set of ethics, and a way of working that are in very close alignment with these commitments.

## 3. Narrative Therapy

When we position the ideas and practices of narrative therapy alongside those in critical suicide studies, we are drawing heavily from the ideas of Michael White [36], David Epston [37], and other practitioners [38] who have built upon their work. A full description of the history and current practice of narrative therapy is beyond the scope of this paper. However, an overview of key narrative ideas and practices, along with their implications for a more critical engagement with suicide, is presented here. With so many possible entry points into understanding narrative therapy, we will start with describing how narrative ideas and practices frame problems.

Simply described, narrative therapy relies on the idea that “problems are manufactured in social, cultural, and political contexts” [39] (p. 27). In turn, a person makes meaning about their life through the stories that are available in those contexts. In other words, problems have an inherent contingency depending on their context, while exerting tremendous influence on a person’s experience of self and identity. Narrative therapy positions self and identity as multiple and fluid, constituted by the available stories we are able to tell about ourselves. These ideas are heavily influenced by Michel Foucault’s thinking on subjectification and the ways in which society and disciplines like psychiatry create standards to measure up to, while “classifying, judging, and determining what is a desirable, appropriate, or acceptable way of life” [39] (p. 8).

Narrative ideas can be productively put to work to engage with the problem of suicide. For example, within a Euro-Western social, cultural, and political context, we can notice the ways in which suicide becomes indexed with psychopathology, where the only available interventions are bio-medical and act on the individual. Certain stories become available from this framing of suicide, including stories about where the problem resides (inside people), and stories about what are allowable responses to the problem (containment in a hospital). Arguably, this framing of suicide has effects on how a person identified as suicidal makes meaning with their life and identity, through the stories that are available for them to tell. The weight of a singular identity as a “suicidal person” appears inescapable, totalizing, with a further shutting down of possibilities for living.

Narrative therapy offers a unique, strengths-based approach for broadening the number of stories available about the problem of suicide. This in turn creates the potential for a person to exercise agency and experience themselves in preferred ways, that go beyond the thin description of “suicidal person”. Narrative therapy serves to expand the possibilities of identity for a person or community in relation to suicide, broadens the ways in which we might understand suicide and its effects upon the lives of people and communities, and offers a praxis for how we might talk with people about suicide and how we might collectively respond. The philosophy, ethics, ways of working in narrative therapy, and their implications for more critical engagements with suicide are described in more detail below.

Narrative therapy positions people as the experts of their own lives, as separate from the problems they are experiencing, and as uniquely qualified to co-research the effects of the problems they are experiencing alongside a narrative practitioner. For example, a person experiencing the effects of suicide would be positioned as a collaborator in responding to the problem, rather than just a person to be subjected to bio-medical interventions, with the narrative practitioner de-centered but still influential. Narrative practitioners pay particular attention to the problem-saturated story [40] which is the dominant story exerting a strong influence on the person’s identity (e.g., the suicidal person).

The practitioner also engages in a form of “double-listening” which enables them to hear the problem story while at the same time, attend to and elicit unique outcomes and discrepancies that point to a departure from the problem. There is often an emphasis on the unique skills and abilities of the person experiencing the problem, reinforcing their agency, and creating the potential to step into a alternative story and identity. Attending to the “absent but implicit” helps to reveal unspoken or hidden knowledges or meanings that could reveal preferences or hopes, which are of particular relevance when engaging with suicide. Narrative ideas and practices suggest a person can never be completely discursively tamed by a problem like suicide, because there are always cracks, openings, and opportunities for alternative stories, positionings, and identities [41].

There are a number of additional practices used throughout narrative therapy that serve to destabilize taken-for-granted assumptions about suicide and create the potential for new responses to suicide. Well-crafted narrative questions can interrogate the problem, recruit in additional perspectives on the problem, strengthen preferred stories, and help the person to align against the problem and with a preferred future. These types of therapeutic questions can help to externalize the problem, separate from the person, within a social, cultural, and political context, creating the needed space for a person to experience agency. Finally, the convening of outsider-witnessing groups, where a group of people are invited to listen to and respond to a narrative interview with a person or another group of people, lends a powerful means to recruit a community of support, generate and amplify new storylines, and mobilize localized knowledge in response to problems. The linking of lives through shared purposes can contribute toward social justice.

## 4. Stepping towards New Possibilities

Through a series of workshops that we have facilitated with mental health and social care providers, we have recently had the opportunity to explore and enact narrative therapy’s philosophy, ethics, and ways of working in a number of community settings to support a more critical engagement with suicide prevention. While not part of a formal research study, we account for some of our early observations here.

For example, in one of these workshops we drew on narrative ideas and practices by inviting participants to engage in a collective learning process about suicide and suicide prevention practices. Specifically, we sought to develop a thicker story about suicide and suicide prevention work, with the ultimate goal of supporting a more complex and generative accounting of participants’ relationship to suicide in the context of their practice as clinicians, youth workers, and community developers. One of us (Jonathan) posed a series of narrative questions to three workshop participants who volunteered to explore these ideas in the presence of the broader group, while the other (Jennifer) invited the remaining workshop participants to engage in a reflective conversation about what they had just witnessed. This led to a much more complicated, deeper, and multi-layered conversation about working with the issue of suicide. For example, participants were able to identify some of the ways that they adhere to the organizational and professional rules as a way to comply with institutional expectations (e.g., performing a suicide risk assessment based on pre-determined questions; documenting levels of risk according to risk categories), while at the same time recognizing some of the limits to this approach. Participants were able to identify a distinct shift in their approach when the topic of suicide came up in their therapeutic conversations, shifting from a narrative, relational, creative, and collaborative way of working together towards a more standardized, bureaucratic approach that was often driven by fear and anxiety. They were also able to see and articulate more clearly some of the understandings that were emerging regarding suicide as a response to unliveable conditions, which in turn opened up possibilities beyond standard suicide risk assessments, to include strategies for mobilizing collective knowledge and social action to support connection, belonging and human flourishing for all.

### 4.1. Multiple Storylines and Discourses

Narrative therapy practices hold promise in scaffolding richer and more expansive ways of understanding and engaging with suicide. Interviewing participants in the presence of others, and re-interviewing witnesses to add additional layers of meaning and possibility, has proven to be an extremely rich way to convene extraordinary and atypical conversations about the problem of suicide, expanding upon the traditional didactic approach of suicide prevention education. Workshop participants described how the enactment of narrative ideas and practices broadened the dialogue, making more things sayable, while surfacing a number of challenges in responding to suicide within contemporary mental health care settings.

For example, through the use of outsider-witnessing groups and narrative questioning, we listened to accounts describing how the problem of suicide not only creates fixed identities among people experiencing distress (i.e., the suicidal person), but also creates fixed identities among helping professionals (i.e., the risk manager), where actions become limited and certain things become unsayable. Through the use of narrative questions designed to elicit a broader story about suicide from the perspective of mental health professionals, it seems that workplace policies and procedures can create a risk management imperative that ultimately governs (and limits) the relationship with the person experiencing distress.

Throughout our workshops, there were rich and consistent descriptions of how risk assessment instruments mediated the therapeutic relationship, with occasions of care appearing to be reduced to tick-boxes and suicide risk factors in a troubling one-size-fits-all mode of practice [32]. It is important to emphasize the fact a number of practices available in some clinical settings are helpful, and no doubt life-saving, in responding to suicide (e.g., dialectical behavioural therapy). However, it was apparent that the healthcare professionals we met with wanted to inhabit multiple storylines where they could be helpful, meet standards of care, uphold their employment obligations, and humanize the care they were providing. This way of working is extremely challenging in a service provision context that is dominated by the demand to only practice using sanctioned evidence-based approaches, which typically leaves little room for irreverent, playful, or imaginative responses. We are interested in seeing how narrative therapy might offer up more expansive possibilities for health professionals to inhabit alternative storylines when caring for people in relationship with suicide.

### 4.2. The Craft of Asking Generative Questions

The use of generative and well-crafted narrative questions could be useful in the context of mainstream suicide risk assessment practice. Such questions create important openings to preferred stories and generate experience [21,38]. This is in contrast to standard suicide risk assessment which are predicated on gathering information. For example, standard practices focus on the specificity of a person’s plan to kill themselves, the likelihood of that plan will result in death, the availability of the means to enact that plan, and the availability of help. The resulting questions tend to create a thin, deficit-based description of the life being experienced by the person in distress. We wonder what would become possible if narrative questions, with their tendency to broaden perspectives, develop preferred storylines, invite fresh thinking, and separate problems from people, became more widely used in response to people in relationship with suicide. Instead of asking how a person plans to kill themselves, we wonder about the life promoting potential of the following questions:
What do these suicidal thoughts say about what you treasure or what might these suicidal thoughts be a protest against [21]?Do these suicidal thoughts take you closer or further away from the values you hold closest?Have you noticed when these suicidal thoughts have the upper-hand? Are there occasions where you have the upper hand?Do you think the description of yourself as a suicidal person is a final story about yourself?


Could such questions elicit a joint understanding of the values a person steps toward—or away from—when contemplating death? Might it enable us to see more clearly some of the potential, unintended harms of preventing suicide at any cost, which often rely on containment, control, and surveillance, especially in acute hospital settings? Could conversations about suicide in mental health or community settings invite more hope and fresh possibilities for living, rather than reproducing predictable and stale conversations that are driven by the prevention imperative to save a life at any cost? We plan to continue to explore how narrative therapy ethics, ideas, and practices might be useful in this precarious and tricky space of practice.

### 4.3. Relational Ethics for Narrative Practice

Narrative and relational approaches to ethics represent a distinct departure from principle-based, expert-driven approaches, such as those espoused by many western moral theories and codified in professional ethics statements [23,27,39]. Narrative therapy practices are predicated on the idea that the person or group seeking help already has an abundance of knowledge, skill, and resources to bring to bear on the problem and that this knowledge can be mobilized through a “de-centred yet influential” [40] practitioner who works in a collaborative, conversational, and creative way to assist the person or group to live into their preferred future.

A relational and narrative approach to ethics is especially well-suited for suicide prevention, since suicide prevention work, like all social care practices [39] is inherently relational, ethical, and fundamentally storied. In this way of working, attention is paid to both the story itself and how it is narrated, which provides clues to how meanings are made within the available cultural resources and dominant discourses. Importantly, identities and ways of going on together are not fixed, and we can re-author our lives in ways that bring us closer to living out our cherished values, commitments, and preferences. There is a clear recognition of our inter-subjective and relational identities which means that both the practitioner and the one seeking help are actively partipating in, and affected by, the emerging knowledge and meaning that is being co-created together. Through mutual exploration, “the ethical narrative that emerges is one that is based on mutuality between, solidarity with, concern for uniqueness or alterity of, care for, and trust in the Other …” [39] (p. 11).

### 4.4. Concluding Comments

Suicide prevention represents a set of shared social practices based on common, and often uninterrogated assumptions that promote a particular view of the problem. When suicide, and our responses to it are understood in exclusively individualistic and scientific terms and are driven by an imperative to prevent it at any cost (i.e., including involuntary hospitalization), we miss the chance to be more creative and compassionate in our responses. It is not that we need to do away with efforts that focus on providing high quality, individual care and treatment to individuals who are suicidal (which may include hospitalization), it is that we need to do more than that if we want to respond to suicide in all its complexity and multiplicity.

As we have argued here, by bringing more attention to the need for socio-politically informed interventions at both the individual and population levels, we believe that practitioners, policy-makers, and community members may be better poised to respond to the complex problem of suicide. We seek to promote a way of doing suicide prevention that delivers on the promise to ensure basic human rights for all citizens. This means recognizing that suicide is sometimes (if not often) a response to policies, systems, and structures that produce vulnerabilities in the form of intergenerational trauma, racism, gender violence, toxic masculinities, social marginalization, and inequities. It also means recognizing our potential for complicity with harm, despite our good intentions as professionals. Practices of narrative therapy are not a panacea for addressing the problem of suicide, but they can offer a powerful analytical framework and set of conversational practices for practicing a critically informed and ethical approach to suicide prevention.

To conclude, we would like to suggest that together, narrative therapy and critical suicide studies can become a creative site of world making. When we move away from a narrow focus on death prevention, towards co-creating a world worth living in, we may be better able to see suicide prevention work as a collective responsibility that is thoroughly ethical in its vision for a more caring, just, relationally engaged, and interdependent future. We hope that this paper can remind us of what is possible when we go beyond the accepted wisdom—which renders suicide as an individual private trouble—and begin to see its multiplicity, historical contingency, and the implications that a narratively informed approach might have for a re-invigorated public conversation about suicide. We see this way of working as hopeful and life-giving for suicidal persons and those who offer care and support, as well as for communities seeking to create the conditions for increased connection, belonging, and ultimately, liveability.

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
