# Peer review of "Re-Thinking Ethics and Politics in Suicide Prevention: Bringing Narrative Ideas into Dialogue with Critical Suicide Studies"

_ijerph, 2019, doi:10.3390/ijerph16183236_

Round 1

Reviewer 1 Report

This is a well written and thought provoking article, building on a good account from the literature. If I could offer some minor critique, I would have liked to see some more explanation of how the ideas that are presented can be carried forward and used by the wider stakeholders that are referenced in the manuscript. I can see the relevance within counselling or more clinical settings, but how can these ideas be applied by the "practitioners, policy- makers and community members" who "may be better poised to respond to the complex problem of suicide." Similarly there is a reference to how responding to suicide is "a collective responsibility? An ethical duty? A relational possibility". Some more guidance on how these notions are realised will be appreciated. There are various reference to ethics in this context, from the title, through various references in the text. Ethics in this context is not really elaborated I would appreciate some more explanation of ethics in this rethinking of suicide.

Author Response

We have responded to the request to provide more detail about how the ideas can be carried forward on p.8 (line 42)

For example, in one of these workshops we drew on narrative ideas and practices by inviting participants to engage in a collective learning process about suicide and suicide prevention practices.  Specifically, we sought to develop a thicker story about suicide and suicide prevention work, with the ultimate goal of supporting a more complex and generative accounting of participants’ relationship to suicide in the context of their practice as clinicians, youth workers, and community developers. One of us (Jonathan) posed a series of narrative questions to three workshop participants who volunteered to explore these ideas in the presence of the broader group, while the other (Jennifer) invited the remaining workshop participants to engage in a reflective conversation about what they had just witnessed.  This led to a much more complicated, deeper and multi-layered conversation about working with the issue of suicide. For example, participants were able to identify some of the ways that they adhere to the organizational and professional rules as a way to comply with institutional expectations (e.g. performing a suicide risk assessment based on pre-determined questions; documenting levels of risk according to risk categories), while at the same time recognizing some of the limits to this approach. Participants were able to identify a distinct shift in their approach when the topic of suicide came up in their therapeutic conversations, shifting from a narrative, relational, creative and collaborative way of working together towards a more standardized, bureaucratic approach that was often driven by fear and anxiety. They were also able to see and articulate more clearly some of the understandings that were emerging regarding suicide as a response to unliveable conditions, which in turn opened up possibilities beyond standard suicide risk assessments, to include strategies for mobilizing collective knowledge and social action to support connection, belonging and human flourishing for all.

We also include more information about our ethical approach on p.10(line 17-36)

4.3. Relational Ethics for Narrative Practice

Narrative and relational approaches to ethics represent a distinct departure from principle-based, expert-driven approaches, such as those espoused by many western moral theories and codified in professional ethics statements [23, 27, 39]. Narrative therapy practices are predicated on the idea that the person or group seeking help already has an abundance of knowledge, skill, and resources to bring to bear on the problem and that this knowledge can be mobilized through a “de-centred yet influential” [40] practitioner who works in a collaborative, conversational, and creative way to assist the person or group to live into their preferred future.

We refined the concluding comments (page 11, line 11)

When we move away from a narrow focus on death prevention, towards co-creating a world worth living in, we may be better able to see suicide prevention work as a collective responsibility that is thoroughly ethical in its vision for a more caring, just, relationally engaged, and interdependent future. We hope that this paper can remind us of what is possible when we go beyond the accepted wisdom – which renders suicide as an individual private trouble – and begin to see its multiplicity, historical contingency, and the implications that a narratively informed approach might have for a re-invigorated public conversation about suicide. We see this way of working as hopeful and life-giving for suicidal persons and those who offer care and support, as well as for communities seeking to create the conditions for increased connection, belonging, and ultimately, liveability.

Reviewer 2 Report

I am afraid this paper should not be published in a journal that publish replicable research.

First, it is not written in the format of an article.

Second, the authors use it to claim several assumptions without empirical or theoretical support, use a non-conventional wording never defined (narrative therapy, critical suicide study), use a judgmental language, and use the paper for self-promotion.

Unsupported judgmental statements include:

Racism (p.2, line 12; p. 3, line 38) Accusation on the current prevention approach and scientific work (“business as usual”, p. 1, line 34; “mainstream suicidologists”, p. 4, line 21).

Besides, the authors repeated several times the same ideas. I do not see how “narrative therapy” can help for suicide prevention.

Please note that in science, personal experience does not count as knowledge.

Finally, it seems that the authors never read Durkheim when they claim that there is an overemphasis on individual/psychological explanations of suicide.

Author Response

We have made our theoretical positioning more clear in the introduction:

We articulate our theoretical position, which is strongly influenced by critical, post-structural, and post-humanist, approaches to scholarly inquiry and praxis. In keeping with a critically reflexive orientation, we write ourselves into the text, unsettling any assumptions about a value-free, neutral or purely objective account of this work.  This is a view from somewhere. We work from the assumption that suicide prevention is a social practice – or assemblage – that comprises bodies, identities, technologies, discourses, institutional artifacts, and cultural practices that are constantly shaping and re-shaping what can be said, thought and done. 

We re-state our intention (p. 4, line 14):

Our intention is to articulate a theoretically grounded, ethically attuned, and practical approach to understanding and responding to suicide. It is not meant to be a panacea, but could be considered as one possibility among others.

We clarify that we do not mean to discredit or devalue others' work (p. 7; line 15):

Nor is it meant to discredit or de-value the important and compassionate work being undertaken by researchers, professional practitioners, volunteers, and family members who are committed to better understanding suicide and who are engaged in the often difficult work of helping another person stay tethered to life. 

Reviewer 3 Report

This paper theorizes about merging practices of narrative therapy with critical suicide studies with the hope of bring new vantage points to suicide prevention practices. The paper is well written and thought provoking. The authors should consider the following: 1. Including examples of “well-crafted narrative questions” and the client’s response to these inquiries would strengthen the paper. Also some examples of the mental health and social care providers’ responses to the workshops would enlighten the reader about impact of discussing the narrative therapy’s philosophy. 2. The authors appeared to have thrown the “baby out with the bath water” in characterizing “care appearing to be reduced to tick-boxes and suicide risk factors in a troubling one-size-fits-all mode of practice”. These statements do not acknowledge that the advances of many of the evidence-based psychotherapies for suicidal clients e.g. DBT have focused on enhancing self-efficacy and have ended the previous approach of labeling suicidal behavior as “manipulative”. In addition, the standardization of suicide prevention practices such as the Zerosuicide approach is intended to be systemic, aspirational and inclusive providing a culture of safe care rather than focusing on the individual client or practitioner.

Author Response

We add more details about narrative questions (p. 10; line 3)

What do these suicidal thoughts say about what you treasure or what might these suicidal thoughts be a protest against [21]? Do these suicidal thoughts take you closer or further away from the values you hold closest? Have you noticed when these suicidal thoughts have the upper-hand? Are there occasions where you have the upper hand? Do you think the description of yourself as a suicidal person is a final story about yourself?

We attempt to bring more nuance to the discussion and hopefully avoid throwing the baby out with the bathwater (p. 9, line 31):

It is important emphasize the fact a number of practices available in some clinical settings are helpful , and no doubt life-saving, in responding to suicide (e.g. dialectical behavioural therapy). 

Also on p.7(line 15):

Nor is it meant to discredit or de-value the important and compassionate work being undertaken by researchers, professional practitioners, volunteers, and family members who are committed to better understanding suicide and who are engaged in the often difficult work of helping another person stay tethered to life. 
